# The Microbiota and Cytokines Correlation between the Jejunum and Colon in Altay Sheep

**DOI:** 10.3390/ani12121564

**Published:** 2022-06-17

**Authors:** Mengjun Ye, Meng Hou, Qimin Peng, Sheng Jia, Bin Peng, Fangfang Yin, Na Li, Jinquan Wang

**Affiliations:** College of Veterinary Medicine, Xinjiang Agricultural University, Urumqi 830052, China; 13668043636@163.com (M.Y.); januaryhoumeng@163.com (M.H.); 15587221846@163.com (Q.P.); jiasheng0519@163.com (S.J.); pengbinwho@163.com (B.P.); wlsa2020@163.com (F.Y.); nali2016vip@163.com (N.L.)

**Keywords:** microbiota, cytokines, Altay sheep, jejunum, colon

## Abstract

**Simple Summary:**

Both the jejunum and the colon secrete unique immune factors that interact with the gut microbiota. Investigating the association of gut microbiota and the host immune system, we detected higher populations of *Bacteroides*, *Fibrobacteres* and *Spirochetes* in the colon than in the jejunum of Altay sheep, which is a unique breed in Xinjiang. Levels of IL-6 and IL-12 were lower in the colon than in the jejunum. IL-10 was positively correlated with *Ruminococcus_2* in the jejunum. These results indicate a potential interaction between intestinal microbiota and the host immune system that may be considered for the prevention of sheep diseases and the screening of probiotics.

**Abstract:**

Both the jejunum and colon release cytokines that interact with intestinal microbiota. However, it is largely unclear which cytokines and microbial populations are involved in the homeostasis of the intestinal ecosystem for sheep health. To address this, we collected contents for isolating microbiota and tissues for determining cytokines from the jejunum and colon of 7-month-old Altay sheep. We used the techniques of 16S rRNA sequencing and ELISA to detect microbial population and cytokine level, respectively. Correlations between microbial population and cytokines were analyzed by Spearman correlation coefficient. The correlation analysis revealed higher populations of *Bacteroides*, *Fibrobacteres* and *Spirochetes* in the colon than in the jejunum, and IL-6 and IL-12 levels were higher in the jejunum than in the colon. Association analysis further revealed a positive association between IL-10 level and both *Ruminococcus_2* and *norank_f_Bifidobacteriaceae* population in the jejunum. The analysis also revealed positive associations between IL-6 level and *Ruminococcaceae_UCG-014* and *Ruminococcaceae_UCG-013* population, IL-10 and *Prevotellaceae_UCG-004*, as well as TNF-α and *Prevotellaceae_UCG-003* in the colon. These results indicate a potential interaction between the intestinal microbiota and the host immune system that needs to be further clarified for considering dietary formulations to maintain animal health and disease prevention.

## 1. Introduction

Microbiota play a pivotal role in modulating host intestinal digestion and immunity [1]. Microbiota populations are unevenly colonized throughout the intestinal system in order to support the functional heterogeneity of distinguishable sections of the intestine [2]. Symbiotic bacteria are able to inhibit pathogenic bacteria that contribute to protecting the intestinal lamina propria from infection [3]. Changes in intestinal bacteria populations may result in host immune response and the release of cytokines into the intestinal tract [4]. Accordingly, advancing our knowledge of intestinal microbiota in association with host cytokines may help maintain healthcare and prevent diseases in animals.

Studies have revealed the relationship between colonic microbiota and host immunity related to colonic diseases [5,6]. However, the relationship between jejunal microbiota and host immunity remains unclear. Studies of sheep’s digestive system have revealed that Firmicutes populate throughout the entire intestinal tract, *Bacteroides* mainly populate in the ileum and large intestine, and Proteobacteria are abundant in the duodenum and jejunum [7]. In small-tailed Han sheep, the abundance of *Bacteroides* gradually increases from the jejunum to the colon [8]. The Altay sheep is a unique breed of Xinjiang and is the main meat-producing sheep [9]. However, the relationship between microbiota and host immunity in Altay sheep is largely unknown. In this study, we investigated host cytokines and microbiota in the jejunum and the colon of Altay sheep. We also studied the potential association of host cytokines with microbial populations in the jejunum and colon.

## 2. Materials and Methods

### 2.1. Animals and Sample Collection

#### 2.1.1. Animals

The high-body-weight (HN) (32.83 ± 2.47 kg) group and the low-body-weight (LN) (23.00 ± 4.33 kg) group of 7-month-old Altay sheep, with 6 per group, were de-wormed and fed with the diet shown in Table 1. By the end of experiments, animals were euthanized by the method approved by the Animal Management and Use Committee of Xinjiang Agricultural University.

#### 2.1.2. Intestinal Content Collection

The gastrointestinal tract was removed from animals right after euthanasia. The content of the jejunum and the colon was collected in sterile tubes and then stored at −80 °C.

#### 2.1.3. Intestinal Tissue Collection

The regions of the jejunum and colon were visually defined, and 8 cm length tissues of each region were sectioned, cleaned in ice-cold PBS (pH 7.36) twice, frozen in liquid nitrogen, and stored at −80 °C.

### 2.2. Microbiota Determination

#### 2.2.1. Bacterial DNA Extraction

Bacterial DNA was extracted from intestinal content using a microbial genomic DNA extraction kit (Tiangen Biochemical Technology Co., Ltd., Beijing, China). DNA quality and quantity were determined using a NanoDrop 2000 spectrophotometer (NanoDrop Technologies, Wilmington, DE, USA). The extracted DNA was stored at −80 °C.

#### 2.2.2. PCR Amplification

The V3-V4 hypervariable regions of the 16S rRNA gene were amplified by the technique of PCR with the primers 338F (5′-ACTCCTACGGGAGGCAGCAG-3′) and 806R (5′-GGACTACHVGGGTWTCTAAT-3′). PCR amplifications were performed in a 20 µL volume, consisting of 10 ng of genomic DNA, 6 μL Master Mix (4 μL of FastPfu Buffer 5 × 2 μL of 2.5 mM dNTPs), 0.8 μL of each primer (5 μM) and RNase Free dH_2_O. Blank controls (no DNA template added to the reaction) were also performed. The PCR reaction was performed using a GeneAmp 9700 thermal cycler (Applied Biosystem, Waltham, MA, USA). Samples were denatured at 95 °C for 3 min, followed by 27 cycles of 95 °C for 30 s, 55 °C for 30 s and 72 °C for 30 s, as well as a final cycle of extension at 72 °C for 10 min. The amplicons were purified with AxyPrep DNA Gel Extraction Kit (Axygen Biosciences, Union City, CA, USA).

#### 2.2.3. 16S Ribosomal RNA (rRNA) Gene Sequencing

Purified amplicons were further prepared using the 16S Metagenomic Sequencing Library Preparation Protocol (Illumina, San Diego, CA, USA) and pooled in equimolar proportion for DNA sequencing in an Illumina MiSeq platform (Illumina, San Diego, CA, USA) and sequenced with 2 × 300-base paired-end reads according to the standard protocols by Majorbio Bio-Pharm Technology Co. Ltd. (Shanghai, China).

### 2.3. Cytokine Determination

For the analysis of cytokines, 0.2 g of intestinal tissues were homogenized with stone pestle in 5 mL ice-cold PBS, followed by incubation on ice for 30–40 min. Tissue extracts were centrifuged at 5000 rpm for 10 min at 4 °C (Eppendorf Centrifuge 5430R). The tissue supernatants were collected for cytokine determination. The concentration of cytokines in 1 mL supernatant was normalized by the weight of the tissue sample.

Levels of interleukins (IL-2, IL-6, IL-10 and IL-12) and tumor necrosis factor-α (TNF-α) were determined using ELISA Kit (SenBeiJia Biological Technology Co., Ltd., Nanjing, China) by comparing the absorbance of samples to the standard curve.

### 2.4. Statistical Analysis

The sheep’s core microbiota was identified by selecting OTUs that were shared by at least 95% of the samples. The Student’s t test was used to analyze microbial diversity indexes, the Shannon index represents the gut microbiota diversity and the coverage index represents the coverage of the sample library [10]. Microbial diversity was determined by the Wilcoxon rank-sum test (the confidence interval was 95%). Significant differences were considered at *p* < 0.05. Statistical evaluation of cytokines was performed by GraphPad Prism 6.01 software (GraphPad Software Inc., La Jolla, CA, USA).

## 3. Results

### 3.1. Intestinal Cytokine Diversities between the Jejunum and Colon

#### The Difference in the Concentration of Intestinal Cytokines

Investigation of intestinal cytokines detected higher levels of IL-6, IL-10, IL-12 and TNF-α in the jejunum than in the colon of the HN sheep, but the level of IL-2 was not significantly different between the jejunum and the colon. On the other hand, we detected higher levels of IL-2, IL-6 and IL-12 in the jejunum than in the colon of the LN sheep, while the levels of IL-10 and TNF-α were not significantly different between the jejunum and the colon (Figure 1). The results revealed a potential association of cytokine profiles with body weights of Altay sheep.

### 3.2. Microbiota Diversities

#### 3.2.1. Sequencing Metrics

After the analysis of the sequences of V3–V4 regions of the bacterial 16S rRNA gene, the Shannon index revealed a divergent microbiota in the jejunum and colon of HN animals (Figure 2a). The coverage index indicated an equality in both the jejunum and the colon (Figure 2b). The total number of identified OTUs was 1160 in the jejunum and 1318 in the colon. Unique OTUs were unevenly distributed along the intestinal tract, with 629 OTUs in the jejunum and 787 OTUs in the colon. The number of unique OUTs was higher in the colon than in the jejunum (Figure 2c).

In LN animals, a higher Shannon index was detected in the colon than in the jejunum (Figure 3a). The coverage index was equal in microbiota populations of the jejunum and colon (Figure 3b). The total number of identified OTUs was 911 in the jejunum and 1273 in the colon. OTUs were distributed in the intestinal tract with 490 OTUs in the jejunum and 852 OTUs in the colon (Figure 3c). Comparing results from the HN and LN groups indicated that microbiota diversity was higher in the colon than in the jejunum regardless of body weight.

#### 3.2.2. Taxonomic Characterization of Microbiota Composition

Studying microbiota composition at the phylum level in the jejunum and colon of the HN animals revealed that Firmicutes Verrucomicrobia, Actinobacteria, Saccharibacteria and Lentisphaerae were the predominant phyla in the jejunum (Figure 4a). Bacteroidetes and Fibrobacteres were the dominant phyla in the colon. Further analysis indicated that the predominant genera were *unclassified_f_Peptostreptococcaceae*, *Paeniclostridium*, *Candidatus_Saccharimonas*, *norank_c_WCHB1-41* and *norank_f_vadinBE97* in the jejunum and *Ruminococcaceae_UCG-010*, *Ruminococcaceae_UCG-005*, *Fibrobacter, Christensenellaceae_R-7_group*, *Rikenellaceae_RC9_gut_group* and *Phocaeicola* in the colon (Figure 4b). Higher populations of *Bacteroides*, *Fibrobacteres*, *Spirochaetae* and *Elusimicrobia* were detected in the colon (Figure 4c).

Studying microbiota composition at the phylum level in the jejunum and colon of the LN animals revealed higher populations of Firmicutes, Proteobacteria, Lentisphaerae and Actinobacteria in the jejunum than in the colon and higher populations of *Bacteroides*, *Spirochaetae* and *Fibrobacteres* in the colon than in the jejunum (Figure 5a). The predominant genera were *Pseudomonas* and *Paeniclostridium* in the jejunum and *Prevotella_1*, *Ruminococcaceae_UCG-010*, *Ruminococcaceae_UCG-005*, *norank_f_Bacteroidales_BS11_gut_group*, *Treponema_2, Christensenellaceae* and *unclassified_f_Lachnospiraceae* in the colon (Figure 5b). The populations of *Bacteroides*, *Spirochaetae* and *Fibrobacteres* were higher in the colon than in the jejunum (Figure 5c).

### 3.3. Correlations between Microbial Populations and Cytokines

To investigate whether microbiota diversity was the causal factor for the divergence in cytokines between the jejunum and colon, we performed Spearman correlation coefficient analysis. As shown in Figure 6a and Appendix A, *Ruminococcus_2* and *norank_f_Bifidobacteriaceae* were positively associated with higher concentrations of IL-10 in the jejunum. In the colon, *norank_f_Clostridiales_vadinBB60_group* was negatively associated with IL-2, and *Ruminococcaceae_UCG-014*, *Ruminococcaceae_UCG-013* and *norank_f_Marinilabiaceae* were positively associated with IL-6. *Prevotellaceae_UCG-004* and *norank_o_Gastranaerophilales* were positively associated with IL-10. *Elusimicrobium* was positively associated with IL-12. *Ruminococcaceae_UCG-005* and *Ruminococcaceae_NK4A214_group* negatively correlated with IL-12. *Norank_f_Bacteroidales_UCG-001* and *Prevotellaceae_UCG-003* were positively correlated with TNF-α, while *Fibrobacter* and *Phoscolarctobacterium* were negatively correlated with it (Figure 6b, Appendix A).

## 4. Discussion

### 4.1. Cytokine Diversities between the Jejunum and Colon

Cytokines such as TNF-α, IL-6 and IL-10 play an important role in the regulation of mucosal immune homeostasis in the digestive tract, and their secretion abnormalities are usually related to diseases [11,12]. Our results showed that that levels of IL-2, IL-6, IL-10 and IL-12 were higher in the jejunum than in the colon. Interleukin is important for the immune system, as it is a transmitter of information and an activator of immune cells [13]. IL-2 is synthesized by T cells after being stimulated by antigens, inducing the expression of immune cells and exerting immune regulation functions, and reducing IL-2 can induce spontaneous colitis in mice [13]. In our study, the higher level of IL-2 determined in the jejunum may explain why the colon is prone to colitis and the jejunum is less inflammatory. IL-6 can induce oxidative stress, and the IL-6 signal on intestinal epithelial cells will promote the differentiation of absorptive epithelium and contribute to the occurrence of intestinal repair pathways in the case of inflammation [14]. Therefore, it can be inferred that the higher IL-6 concentration will help repair the damaged part of the jejunum and ensure the digestion and absorption function. IL-10 is a pleiotropic cytokine. It has immunostimulatory and immunosuppressive effects in a variety of cell tissues. It participates in anti-inflammatory effects in immune responses and is regulated by IL-6. Its main role is to limit inflammation and maintain intestinal immune homeostasis, and IL-10-deficient mice can spontaneously form colorectal cancer when they are infected with intestinal bacteria [15,16]. Above all, higher levels of cytokine in the jejunum indicate that the immune system function was more active in the jejunum than the colon.

### 4.2. Analysis of Microbiota Diversities in the Jejunum and Colon

The intestinal microbiota are called the “second genome” and play an important role in maintaining the growth, development and health [17]. Our results show that Firmicutes was the most abundant phylum in the jejunum and colon of Altay sheep. This is consistent with the results of studies on the intestinal flora of other breeds of sheep [5,18,19] and cattle [20]. Firmicutes in the intestinal tract of ruminants can degrade fiber in food [21] and helps with food digestion. At the same time, it is also the most abundant and representative phylum in the intestine of ruminants [22]. In our study, corn is one of the main components in the diet of sheep, and this may be the reason for the abundance of Firmicutes.

In addition, the dominant flora in the jejunum and colon of Altay sheep were characterized. The Actinobacteria and *Lentisphaerae* were the predominant phyla in the jejunum, while the *Bacteroides* and *Fibrobacteres* were the predominant phyla in the colon. Additionally, the higher populations of *Bacteroides*, *Fibrobacteres* and *Spirobacteria* were detected in the colon. This may due to the fact that *Bacteroides* contributes to the digestion and absorption of complex carbohydrates in diets [23]. In addition, it is regarded as the most important part of the digestive fiber in the intestines of herbivores, which can provide a series of degrading enzyme systems to destroy the plant cell wall [24] and help the digestion of food in the gut [25]. In the study of the intestinal microbiota of Mongolian sheep, a higher abundance of *Bacteroides* was detected in the large intestine than in the small intestine [26]. For cattle, the abundance of *Bacteroides* was also higher in the rumen and reticulum than that in the small intestine [27]. In the jejunum of small-tailed Han sheep [8], the dominant genera were *Ruminococcus*, *Clostridium* and *Acetitomaculum*, and the dominant genera in the large intestine were *Ruminococcus* and *Bacteroides*. In our study, at the genus level, *Paeniclostridium* and *Peptostreptococaceae* were the dominant genera in the jejunum, and *Ruminococcaceae*, *Christensenellaceae* and *Rikenellaceae* were the dominant genera in the colon. The results indicate that the dominant phyla in the gut of different breeds of sheep are similar, but the genera are diverse, which may related to breed, genetics, diet composition and the growth environment.

### 4.3. Correlation of Intestinal Microbiota and Cytokines

In the jejunum and colon of Altay sheep, many genera are correlated with cytokines. The correlation between intestinal microbiota and cytokines has been found in many studies. For example, changes in the TNF-α concentration in patients with ulcerative colitis are correlated with changes in intestinal flora [28]. The IL-10 change in patients with non-alcoholic fatty liver can cause imbalance of intestinal flora [29]. Scholars have observed that the microbiome–cytokine interaction has stimulus specificity and cytokine specificity [30]. At present, there are few studies on the correlation between sheep flora and cytokines. This may due to people pay more attention to the research on nutritional control and absorption function in the jejunum and colon. Above all, the immune function of the microbiota in jejunum and colon need to be further studied.

## 5. Conclusions

The microbiota communities between the jejunum and the colon in Altay sheep are diverse. The immune cytokine concentrations are higher in the jejunum than in the colon. The increase in some microbiota may mediate the increase in IL-6 and other immune factors to influence immune system. These results indicate a potential interaction between intestinal microbiota and the host immune system that needs to be further clarified in order to consider dietary formulations to maintain animal health and disease prevention.

## Figures and Tables

**Figure 1 animals-12-01564-f001:**
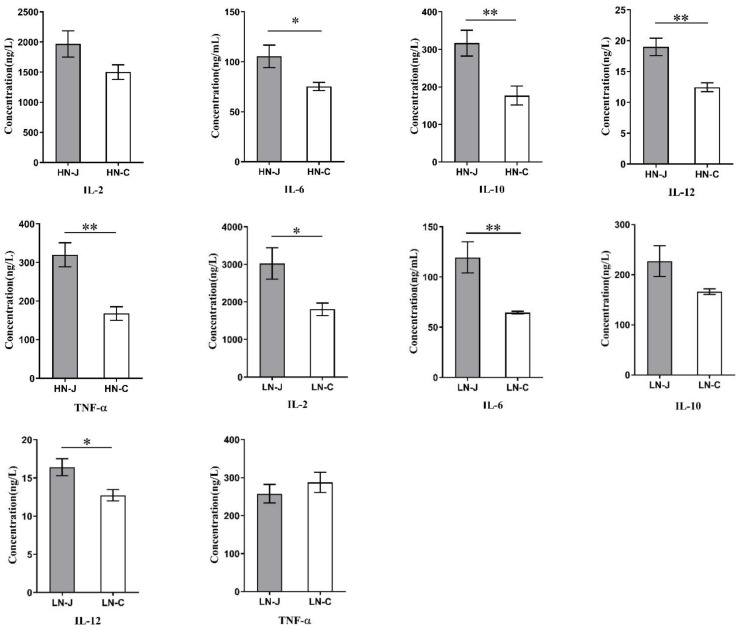
Cytokines in the jejunum and colon of HN and LN. HN-J, the jejunum of HN; HN-C, the colon of HN; LN-J, the jejunum of LN; LN-C, the colon of LN. Columns, mean of sextuplicates; bars, SD. Statistical significance is indicated by * *p* < 0.05, ** *p* < 0.01.

**Figure 2 animals-12-01564-f002:**
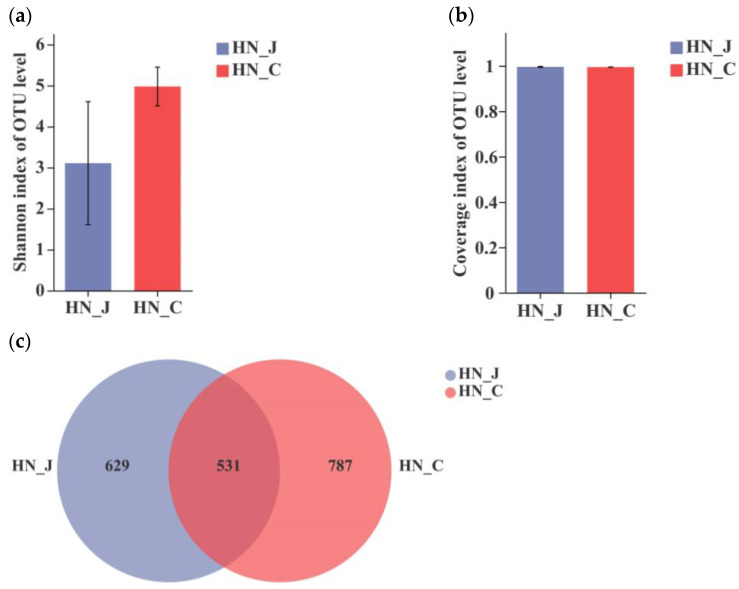
Index diversities in the jejunum and colon of HN. (**a**) Shannon index of the OUT level; (**b**) coverage index of OUT level; (**c**) OTUs Venn diagram. HN-J, the jejunum of HN; HN-C, the colon of HN; columns, mean of sextuplicates; bars, SD.

**Figure 3 animals-12-01564-f003:**
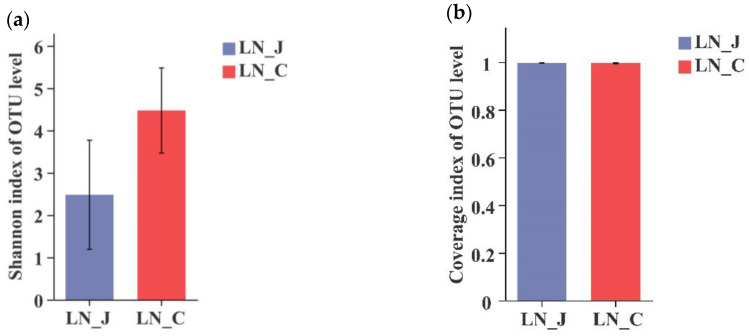
Index diversities in the jejunum and colon of LN. (**a**) Shannon index of OUT level; (**b**) coverage index of OUT level; (**c**) OTUs Venn diagram. LN-J, the jejunum of LN; LN-C, the colon of LN; columns, mean of sextuplicates; bars, SD.

**Figure 4 animals-12-01564-f004:**
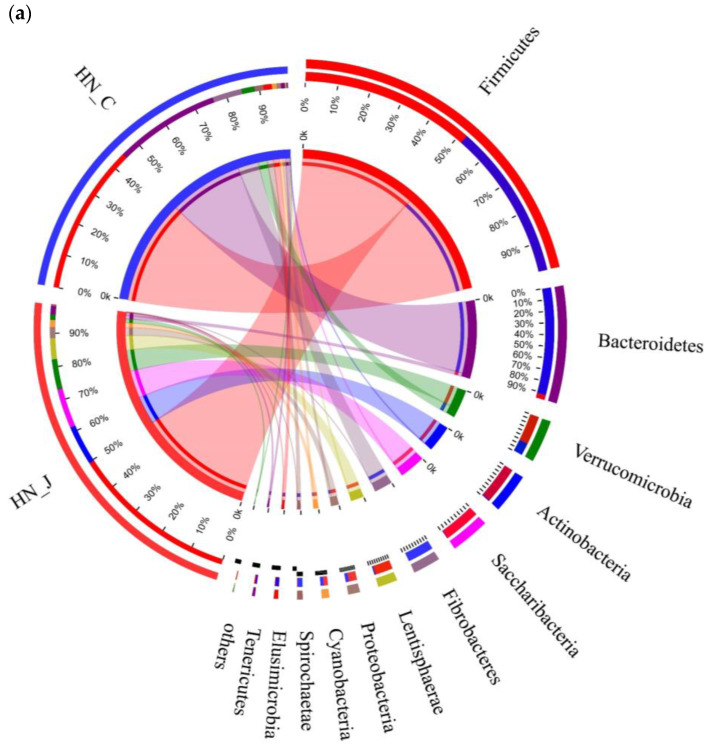
Microbiota diversities in the jejunum and colon of HN. (**a**) Circos diagram (phylum level); (**b**) community compositions (genus level); (**c**) microbial diversities between the jejunum and colon (determined by the Wilcoxon rank-sum test; the confidence interval was 95%). Statistical significance is indicated by * *p* < 0.05.

**Figure 5 animals-12-01564-f005:**
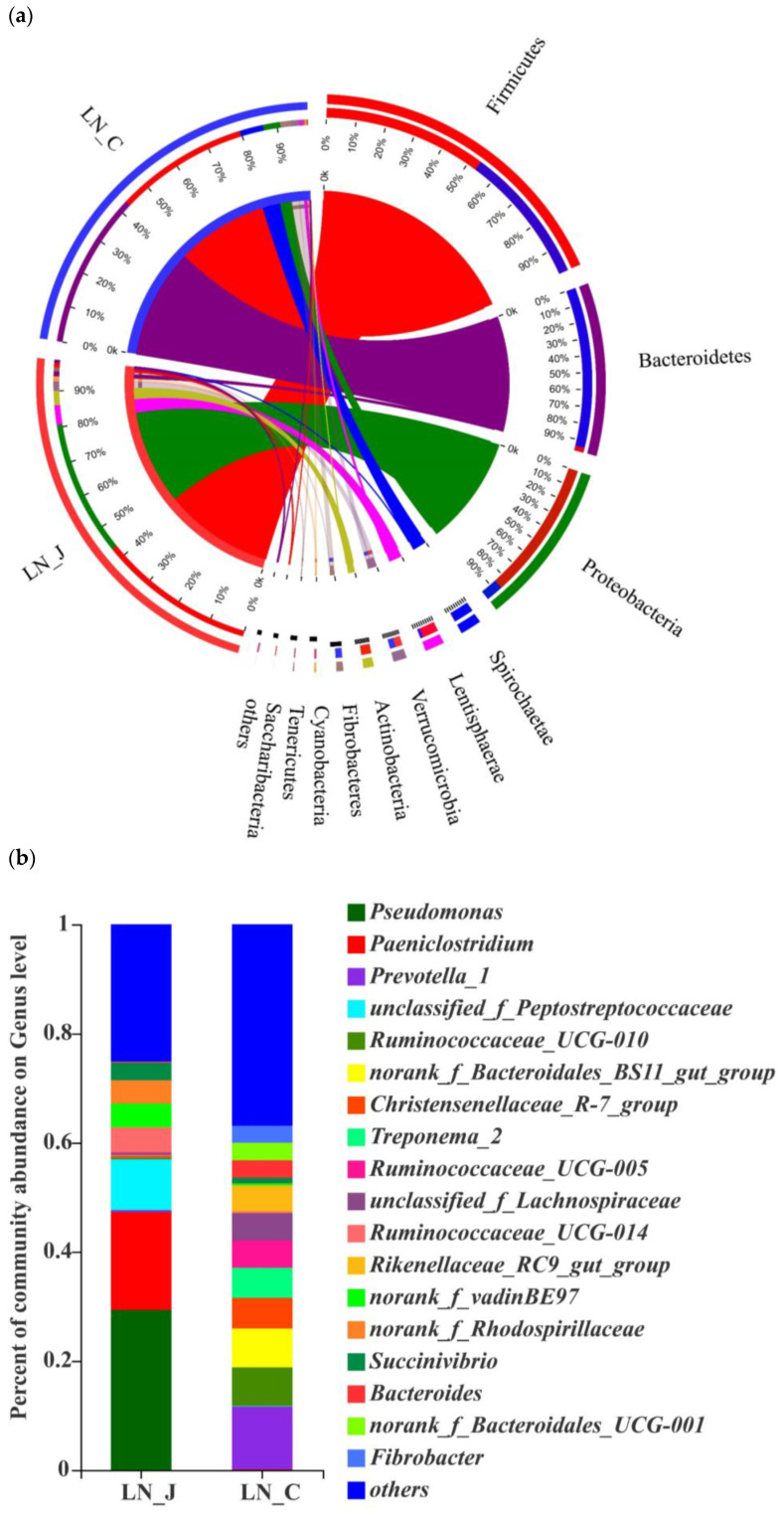
Microbiota diversities in the jejunum and colon of LN. (**a**) Circos diagram (phylum level); (**b**) community compositions (genus level); (**c**) microbial diversities between the jejunum and colon (determined by the Wilcoxon rank-sum test, the confidence interval was 95%). Statistical significance is indicated by * *p* < 0.05.

**Figure 6 animals-12-01564-f006:**
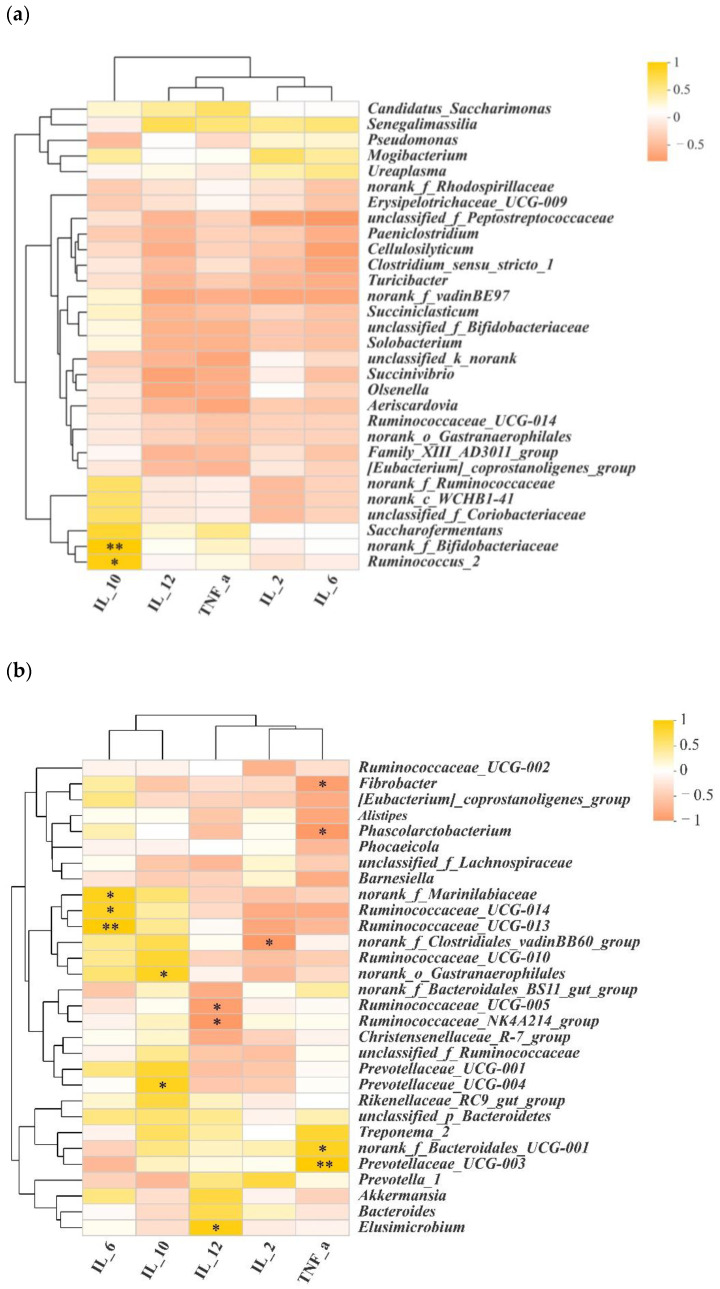
Correlations between microbial populations and cytokines. (**a**) Correlations between microbial population and cytokines in the jejunum (genus level); (**b**) correlations between microbial population and cytokines in the colon (genus level). Statistical significance is indicated by * *p* < 0.05, ** *p* < 0.01.

**Table 1 animals-12-01564-t001:** Diet composition.

Diet Composition (%)	
Corn	29.53
Cottonseed	2.62
Soybean	3.32
Bran	6.56
CaHCO_3_	0.18
Limestone	0.66
NaCl	0.44
Premix	0.44
Cornstalk	56.25

Note: The rations were purchased from Xinjiang Medical University. Premix components: Cu, 200 mg·kg^−^^1^; Fe, 1200 mg·kg^−^^1^; Mn, 200 mg·kg^−^^1^; Zn, 400 mg·kg^−^^1^; I, 20 mg·kg^−^^1^; Se, 15 mg·kg^−^^1^; VA, 80,000 IU·kg^−^^1^; VD3, 20,000 IU·kg^−^^1^; VE, 500 IU·kg^−^^1^.

## Data Availability

Data are available in a publicly accessible repository. The data presented in this study are openly available in NCBI, SUB10590743-Summary/Sequence Read Archive (SRA)/Submission Portal (nih.gov).

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
