# Peer review of "The Microbiota and Cytokines Correlation between the Jejunum and Colon in Altay Sheep"

_animals, 2022, doi:10.3390/ani12121564_

Round 1
Reviewer 1 Report
The manuscript by Ye et al. provided valuable data correlating microbiota and cytokines in the colon versus jejunum, but I did not find any controls used in this study. Microbiome analysis must use positive control, negative control, and process control to avoid bias between kits used to extract DNA and PCR amplification. Please review the article published in FEMS Microbiology Ecology on “Issues and current standards of controls in microbiome research” (https://doi.org/10.1093/femsec/fiz045).
Even though your results look good, the data is invalid without these controls. For other researchers to repeat what you have done, it is necessary for the authors to include the control data, so authors need to address this issue.
Author Response
Dear Sir/Madam,
Thank you for your comments concerning our manuscript entitled “The Microbiota and cytokines correlation between the jejunum and colon in Altay sheep” (Manuscript ID: animals-1738918). Those comments are all valuable and very helpful for revising and improving our paper, as well as the important guiding significance to our researches. We have studied comments carefully and have made correction which we hope meet with approval. The main corrections in the paper and the responds to the reviewer’s comments are as flowing:
Responds to the comments:
- Response to comment: did not find any controls used in this study.
Response: Thanks for the recommendation, in our experiment, all the sheep were healthy animals without any treatment. We have performed Microbiome analysis by using blank controls to avoid bias between kits used to extract DNA and PCR amplification. For this issue, we have mentioned in the revised Materials Methods, as shown in 2.2 Microbiota Determination.

Reviewer 2 Report
Dear Authors,
Thank you for submitting this interesting paper describing the difference in cytokines and microbiome diversity in the jejunum and colon of sheep. I found the paper to be engaging overall.
At current however, there seem to be some large revisions required in the manuscript to ensure the work is scientifically robust. I have attached the PDF version of the manuscript with specific comments. Additionally, please consider the following points:
- Methods. Ensure that all aspects of the methods (number of animals killed, diet of animals) is covered in full.
- Data analysis. It is often unclear what tests were used. Please state all statistical tests (and diversity indices) in your data analysis section.
- Use of tests. When stating a p value, please provide the test statistic (e.g. r value or t value) alongside the p value. Please provide the actual p value to 3 decimal places).
- Citations. Please ensure that the references and citations are formatted correctly to Animals as currently they are not.
- Implications. From reading the paper, the wider reasoning for this work is unclear. Please provide a more careful explanation as to why this research is necessary, along with future directions as a result.

Author Response
Dear Sir/Madam,
Thank you for your comments concerning our manuscript entitled “The Microbiota and cytokines correlation between the jejunum and colon in Altay sheep” (Manuscript ID: animals-1738918). Those comments are all valuable and very helpful for revising and improving our paper, as well as the important guiding significance to our researches. We have studied comments carefully and have made correction which we hope meet with approval. The main corrections in the paper and the responds to the reviewer’s comments are as flowing:
Responds to the comments:
- Response to comment: Extensive editing of English language and style required.
Response: After careful consideration of your suggestion, we have invited a native English-speaking professor to revise the language structure and grammar of the article.
- Response to comment: Methods. Ensure that all aspects of the methods (number of animals killed, diet of animals) are covered in full.
Response: According to your suggestion, we rewrite the material method in detail and added the diet composition table in the section 2.1.1.
- Response to comment: Data analysis. It is often unclear what tests were used. Please state all statistical tests (and diversity indices) in your data analysis section.
Response: In response to this issue, we have rewritten the analysis methods for data in Section 2.4.
- Response to comment: Use of tests. When stating a p value, please provide the test statistic (e.g. r value or t value) alongside the p value. Please provide the actual p value to 3 decimal places).
Response: Since we use column charts in most cases to represent whether the differences are significant or not, we re-write the figure legends in detail. Additionally, for the R-values and P-values in the heatmap, we added the relevant material in the supplement tables.
- Response to comment: Citations. Please ensure that the references and citations are formatted correctly to Animals as currently they are not.
Response: We have revised the format of the references.
- Response to comment: Implications. From reading the paper, the wider reasoning for this work is unclear. Please provide a more careful explanation as to why this research is necessary, along with future directions as a result.
Response: We reorganized the simple summary, abstract and introduction to make the article more readable and highlights the necessity of this study.
Yours,
Mengjun Ye

Reviewer 3 Report
The authors conducted interesting research on microbiota and cytokines.
However, the publication requires a few corrections.
no Simple Summary
References must be numbered in order of appearance in the text. And the authors did not. The publication includes surnames and initials of names, not references to numbers.
did the authors have the consent of the animal ethics committee to conduct the research?
L31remove the period in front of the parenthesis
L70 We collected the contents of the jejunum and colon and intestinal tissue of each - how much, all, part was taken or was it chilled at once?
Sample collection and processing - I propose to divide into two separate sections for microbiota and cytokines
L89 The intestinal tissue is processed by low-temperature manual grinding. - what temperature? please specify; was the tissue pre-cleaned, somehow prepared?
L65/66 up. All of them were fed indoor under the same 65
condition with the same diet for one month. - please provide animal nutrition tables. has the feed been microbiologically tested before? is the sanitary condition determined?
the publication requires many corrections, especially in the research methodology.
Author Response
Dear Sir/Madam,
Thank you for your comments concerning our manuscript entitled “The Microbiota and cytokines correlation between the jejunum and colon in Altay sheep” (Manuscript ID: animals-1738918). Those comments are all valuable and very helpful for revising and improving our paper, as well as the important guiding significance to our researches. We have studied comments carefully and have made correction which we hope meet with approval. The main corrections in the paper and the responds to the reviewer’s comments are as flowing:
Responds to the comments:
- Response to comment: no Simple Summary
Response: In the revised manuscript, a simple summary has been added.
- Response to comment: References must be numbered in order of appearance in the text. And the authors did not. The publication includes surnames and initials of names, not references to numbers.
Response: We have revised the format of the references as Animals required.
- Response to comment: the consent of the animal ethics committee
Response: We have the consent of the animal ethics committee to conduct the research. This section is mentioned in revised Materials and Methods.
- Response to comment: contents of the jejunum and colon and intestinal tissue of each - how much, all, part was taken or was it chilled at once?
Response: We have rearranged and re-written the Materials Methods with a detailed description of the process and volume of sampling.
- Response to comment: Sample collection and processing
Response: We have followed your suggestion to divide sample collection and processing into two separate sections for microbiota and cytokines.
- Response to comment: The intestinal tissue is processed by low-temperature manual grinding. - what temperature? please specify; was the tissue pre-cleaned, somehow prepared?
Response: We have added conditions in revised Materials and Methods. “For the analysis of cytokines, 0.2 g of intestinal tissues were homogenized with stone pestle in 5 mL ice-cold PBS, followed by incubation on ice for 30-40 min. Tissue extracts were centrifuged at 5000 rpm for 10 min at 4 ◦C (Eppendorf Centrifuge 5430R). The tissue supernatants were collected for cytokine determination. The concentration of cytokines in 1 ml supernatant was normalized by the weight of the tissue sample.”
- Response to comment: please provide animal nutrition tables. has the feed been microbiologically tested before? is the sanitary condition determined?
Response: The animal nutrition table has been provide in the revised manuscript. Feed was purchased directly, and all animals were fed the same feed, so we did not measure the microbial content of the feed.
Yours,
Mengjun Ye

Round 2
Reviewer 1 Report
Thanks for incorporating details.
Reviewer 2 Report
Dear Authors,
Many thanks for submitting this revised version of the manuscript for review. You have taken into account the feedback provided on the initial review of the paper. You have also shown clearly where changes have been made to the work, as shown with the tracked changes sections of text. The developments to the manuscript have resulted in a more robust paper overall. In light of the revisions, the paper is now in a much better position for consideration.
Reviewer 3 Report
the publication has been corrected in accordance with the comments of the reviewer. I have no more comments. Publications can be published